computational biology, immunology, microbiology

interferon, innate immunity, ring vaccination, partial differential equations, cellular automata

**Author for correspondence:**
Ruian Ke
e-mail: rke@lanl.gov

# Autocrine and paracrine interferon signalling as 'ring vaccination' and 'contact tracing' strategies to suppress virus infection in a host

G. Michael Lavigne[1], Hayley Russell[1], Barbara Sherry[2] and Ruian Ke[1,3]

[1]Department of Mathematics, and [2]School of Veterinary Medicine, North Carolina State University, Raleigh, NC 27606, USA
[3]T-6, Theoretical Biology and Biophysics, Los Alamos National Laboratory, Los Alamos, NM 87545, USA

 RK, 0000-0001-5307-8934

The innate immune response, particularly the interferon response, represents a first line of defence against viral infections. The interferon molecules produced from infected cells act through autocrine and paracrine signalling to turn host cells into an antiviral state. Although the molecular mechanisms of IFN signalling have been well characterized, how the interferon response collectively contribute to the regulation of host cells to stop or suppress viral infection during early infection remain unclear. Here, we use mathematical models to delineate the roles of the autocrine and the paracrine signalling, and show that their impacts on viral spread are dependent on how infection proceeds. In particular, we found that when infection is well-mixed, the paracrine signalling is not as effective; by contrast, when infection spreads in a spatial manner, a likely scenario during initial infection in tissue, the paracrine signalling can impede the spread of infection by decreasing the number of susceptible cells close to the site of infection. Furthermore, we argue that the interferon response can be seen as a parallel to population-level epidemic prevention strategies such as 'contact tracing' or 'ring vaccination'. Thus, our results here may have implications for the outbreak control at the population scale more broadly.

## 1. Introduction

The innate immune response provides critical protection against pathogen invasion of humans and other animals prior to establishment of adaptive immunity. It relies on multiple cytokines, chief among them being interferons (IFNs), a large, diverse family of signalling proteins that together induce a protective response [1]. The importance of IFN in the defence against viral infections is demonstrated by the fact that essentially all viral pathogens have developed mechanisms to interfere with or suppress the host IFN response [2–4]. Indeed, viral evasion of the IFN response strongly determines the rate of viral replication, the success of transmission and infection establishment in new hosts [5] and the range of species infected [6]. The capacity to inhibit the IFN response determines species tropism for human immunodeficiency virus [7], dengue virus [8], rotavirus [9], measles virus [10] and influenza virus [11]. Interestingly, multiple lines of recent evidence show that severe symptoms and life-threatening disease from SARS-CoV-2 infection is linked to inhibition of IFN signalling or inborn deficiency in IFN immunity [12–14].

The IFN response is commonly described by its two components: first, viral induction of IFN, and second, IFN induction of antiviral genes [15]. Upon infection, viral RNAs or DNAs are detected by the cell, triggering a signalling cascade that results in the production of Type I IFNs [16,17]. These IFN molecules are then secreted and bind to surface receptors located on the cell membrane. IFN binding to the surface of the cell from which it is produced is referred to as autocrine signalling, whereas binding to the surface of any other cell is referred to as

**2**

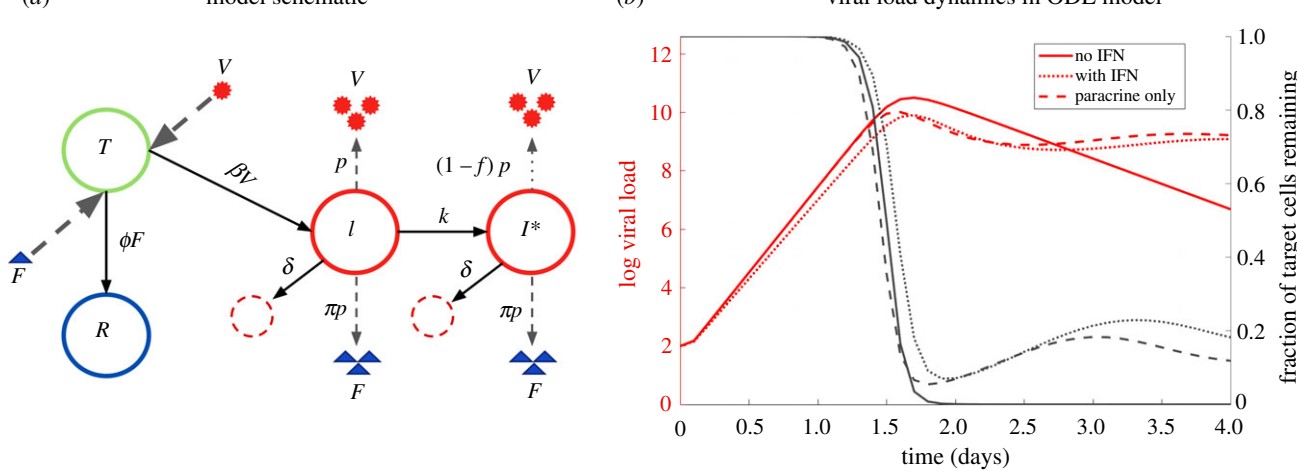

**Figure 1.** Schematic diagram of the viral infection dynamics and the IFN response and simulations of the corresponding ODE model. (*a*) Schematic diagram with parameters in the model. Solid arrows indicate transition of cells from one state to another; dashed arrows indicate the production or binding of viruses and IFNs from cells. (*b*) Simulations of ODE model without IFN, with IFN, and with only paracrine signalling demonstrate the effects of different parameter regimes on the growth of viral load. Simulation of a model with the paracrine signalling alone ($k = 0$, dashed lines) shows no notable effect on the initial exponential growth rate compared to the no-IFN case ($k = 0$ and $\pi = 0$, solid lines), while both autocrine and paracrine signalling together can slow the growth of viral load (dotted lines). (Online version in colour.)

paracrine signalling. This binding initiates a series of signalling events that ultimately result in the production of interferon-stimulated genes (ISGs), the expression of which repress viral replication in the cell at multiple steps [18]. In an uninfected cell, binding of IFN to its receptor and subsequent IFN signalling renders the cell refractory to viral infection, while in an infected cell, this signalling can suppress viral replication and decrease release of viral progeny from the cell. An elegant analysis of the virus-induced IFN response at the single cell level demonstrated that paracrine signalling early in infection shapes the overall IFN response [19]. However, the inflammatory response elicited by IFN can have deleterious effects on the host if uncontrolled [13,20,21].

Although the molecular mechanisms of IFN signalling have been well characterized, the systems-level properties arising from the individual host-cell response, particularly how the host cells collectively stop viral infection at the site of exposure before the infection becomes systemic, remain unclear. To address these questions, we use modelling approaches to understand how IFN signalling can stop early infection (e.g. at the site of initial entry) before adaptive immunity is developed. Previous modelling of virus infection and the IFN response has focused on the role of IFN response after the infection becomes systemic and used ordinary differential equations (ODEs) [22–25]. ODE models necessarily include the implicit assumption that the host is treated as a single well-mixed compartment, and thus they neglect the spatial structure of infection. Influenza infection, for example, starts at the epithelial lining of the upper respiratory tract, which is an inherently a spatial process [26]. Therefore, to investigate the interaction between virus and the IFN response early in infection, a spatially explicit model is most appropriate. We previously modelled the spread of virus infection and the effectiveness and robustness of IFN signalling among host cells using a network approach [27,28]. However, the explicit roles of autocrine and paracrine signalling in suppressing virus spread are not clear.

Here, we develop various models with or without explicitly considering the impact of the spatial arrangement of cells to examine the roles of autocrine and paracrine IFN signalling. We first show that, in well-mixed ODE models, autocrine signalling can impact the course of infection by inhibiting virus

production from already infected cells, whereas paracrine signalling has negligible impact on the growth of viral load during early infection when target cells are abundant. By contrast, in models explicitly considering spatial spread, IFN paracrine signalling can stop viral infection by segregating susceptible cells from areas of infection with an insulating layer of protected cells. This strategy parallels the control strategies of 'ring vaccination' and 'contact tracing' in epidemiology and outbreak control which aim to stop spread of infection by targeting the most at-risk individuals [29–31].

## 2. Methods

### (a) A non-spatial model of well-mixed viral infection

We first develop a model of viral infection with IFN signalling using ODEs. In this approach, we assume that cells, viruses and IFN are well mixed and thus spatial structure is not considered. Such models have been well established by previous work on *in vivo* models of virus–immune interaction during systemic infection [22,24]. The equations of our model are as follows:

$$
\begin{aligned}
\frac{dT}{dt} &= -\beta VT - \phi FT + \rho R \\
\frac{dI}{dt} &= \beta VT - \delta I - kI - \phi FI \\
\frac{dI^*}{dt} &= kI + \phi FI - \delta I^* \\
\frac{dR}{dt} &= \phi FT - \rho R \\
\frac{dV}{dt} &= pI + (1-f)pI^* - cV \\
\frac{dF}{dt} &= \pi p(I + I^*) - cF.
\end{aligned}
\tag{2.1}
$$

In this model (see figure 1*a* for a schematic), cells are categorized into one of four states: uninfected target cells $T$, productively infected cells $I$, infected cells that are in an antiviral state $I^*$ and refractory cells $R$. Uninfected cells are infected by virions $V$ at rate $\beta$ or become refractory to infection through paracrine signalling of IFN ($F$) at rate $\phi$. Binding of IFN molecules to IFN receptors on infected cells ($I$), including both autocrine and paracrine IFN signalling, may trigger an antiviral response in those cells, such that virus production is

inhibited or reduced [5]. We model the impacts of autocrine and paracrine signalling using two separate terms (i.e. $kI$ and $\phi FI$). Note that $F$ in our model represents the ambient concentration of unbound IFN (under the assumption of homogeneous concentration of IFN). We assume that autocrine signalling occurs independent of the ambient IFN concentration, because once produced from infected cells, IFNs preferentially bind to the producing cell due to proximity. The transition towards an antiviral state due to autocrine signalling is thus modelled by $kI$ (i.e. independent of ambient IFN concentration). By contrast, the rate of transition due to the paracrine signalling is modelled to be dependent on the IFN concentration with the term $\phi FI$.

We assume that infected cells (both $I$ and $I^*$) die at the same *per capita* rate $\delta$. Refractory cells remain protected for an average time of $1/\rho$ before returning to the susceptible state (i.e. becoming target cells again). Infected cells, $I$, release viruses at rate $p$, whereas infected cells at an antiviral state, $I^*$, release virions at a reduced rate $(1-f)p$, where $f$ is the fraction of reduction. For simplicity, we further assume that both $I$ and $I^*$ cells release IFNs at rate $\pi p$ and that viruses and IFNs are cleared at *per capita* rate $c$. Note that since the time scale of the dynamics of IFNs is much faster than the time scale of dynamics of the cells, we can make the quasi-equilibrium assumption for the concentration of IFN and then the level of IFNs are related to infected cells as $F = (\pi/c)p(I+I^*)$. Therefore, if IFN is cleared in the system at a rate different from $c$, the level of IFN can be compensated in the system by changing the value of $\pi$.

## (b) A spatial model of viral infection with IFN signalling

We next develop a partial differential equation (PDE) model of viral infection and IFN response. This model explicitly considers the spatial arrangement of cells, virions, and IFNs, thus more accurately representing the dynamics of infection in an epithelial tissue. We assume that susceptible cells $T$ are arranged on a one-dimensional space with spatial variable $x \in [0, L]$ with a uniform initial density $T_0$. Viruses and IFNs can diffuse to nearby locations, in contrast to the ODE model where viruses and IFNs are assumed to instantaneously be evenly distributed once produced. Virions and IFNs diffuse across the spatial domain with diffusion coefficients $D_V$ and $D_F$, where we take $D_F \gg D_V$ since IFNs are much smaller than virions and therefore diffuse at a much greater rate [32,33]. These diffusion parameters determine the characteristic length scales on which IFNs and virions will be active [34]. The initial conditions are taken to be such that the domain is populated only with target cells at a constant density and a single infected cell at the position $x = 0$, which is achieved using a Dirac delta distribution $\delta_0(x)$. The boundary conditions are taken to be homogeneous Neumann at $x = 0$ to represent reflective symmetry of the spread of infection, and homogeneous Dirichlet at far-field $x = L$. The equations of the model are as follows:

$$\frac{\partial T}{\partial t} = -\beta VT - \phi FT + \rho R$$

$$\frac{\partial I}{\partial t} = \beta VT - \delta I - kI - \phi FI$$

$$\frac{\partial I^*}{\partial t} = kI + \phi FI - \delta I^*$$

$$\frac{\partial R}{\partial t} = \phi FT - \rho R$$

$$\frac{\partial V}{\partial t} = pI + (1-f)pI^* - cV + D_V \frac{\partial^2 V}{\partial x^2}$$

$$\frac{\partial F}{\partial t} = \pi p(I+I^*) - cF + D_F \frac{\partial^2 F}{\partial x^2}$$

boundary conditions: $\frac{\partial F}{\partial x} = \frac{\partial V}{\partial x} = 0$ at $x = 0$, $F = V = 0$ at $x = L$

initial conditions: $T(x,0) = T_0$, $I(x,0) = \delta_0(x)$.

(2.2)

## (c) A cellular automata model

We lastly develop a 2D Cellular Automata (CA) model to model the spatial progression of viral infection with IFN signalling. The CA framework allows us to consider the spatial infection spread governed by a stochastic process. By developing a CA model, we can more accurately depict the nature of early viral infection in epithelial tissue

In our CA model, each individual epithelial cell is tracked explicitly as a grid point on a stationary $N \times N$ lattice. Cells interact locally with other cells near to themselves based on predefined rules for the production and diffusion of virions and IFN particles. A cell can be in any of five states: healthy, exposed, productively infected, protected, or dead. The CA is initialized with a single infected cell located at the centre of the grid of otherwise healthy target cells. Furthermore, virion and interferon particles are not explicitly modelled agents, but rather we consider their production, diffusion and binding to recipient cells to occur within the duration of a single iteration of the CA. This choice allows us to take large time steps and is less costly than explicitly modelling the random walk of each particle. More detailed specifications of the CA model can be found in electronic supplementary material.

## 3. Results

## (a) The roles of autocrine and paracrine IFN signalling in a non-spatial well-mixed infection

We first constructed a model (see figure 1$a$ for a schematic) and analysed the roles of autocrine and paracrine IFN signalling using ordinary differential equations (ODEs) (see Methods). To understand the impacts of autocrine and paracrine signalling on the virus dynamics after initial viral exposure, we calculated the basic reproductive number $R_0$ of the virus using the next generation matrix technique [35]. Note that $R_0 = 1$ is the threshold for establishment of infection, and viral population only grows when $R_0 > 1$. Thus, for an effective innate immune response to halt viral infection, $R_0$ has to be less than 1. For the above model, we find

$$R_0 = T_0 \frac{\beta p}{c\delta} \left(1 - \frac{fk}{\delta + k}\right).$$

This expression shows that the reduction of $R_0$ due to autocrine signalling is $fk/(\delta + k)$, where $f$ is the inhibition of virus production due to the cellular antiviral response and $k/(\delta + k)$ is the probability that an infected cell becomes antiviral by the autocrine pathway before cell death occurs. *In vitro* experiments suggest that the fraction of infected cells that successfully enter an antiviral state is in general low [36–38], i.e. $k/(\delta + k)$ is much less than 1. If this observation is consistent with IFN response *in vivo*, then our results suggest that autocrine signalling has limited impact on stopping viral infection during initial stage of infection.

Importantly, we found that the parameters governing paracrine IFN signalling (i.e. $\phi$, $\pi$) do not appear in the expression for $R_0$ (i.e. paracrine signalling alone does not change the infection threshold). Therefore, the ODE model makes the surprising prediction that when cells, viruses and IFN are well mixed (as assumed in our ODE model and other models [22,24]), paracrine signalling has a negligible role in halting infection during early infection when the number of target cells are abundant. We further performed simulations of the model (see electronic

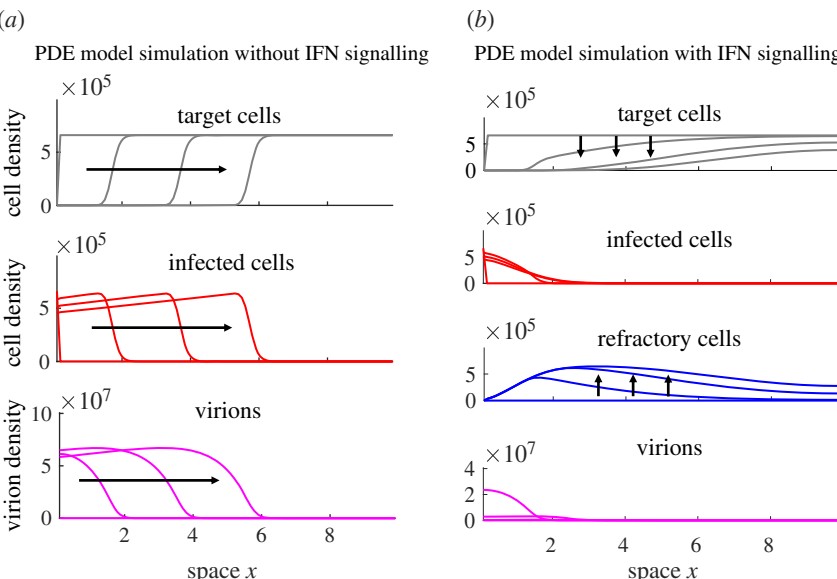

**Figure 2.** Model simulations show that paracrine IFN signalling strongly interferes with the spatial spread of infection. Shown is the solution of the model system at successive time points, with arrows indicating the direction of progression with time. (*a*) A representative simulation of the PDE model with $k = \pi = 0$ (no IFN), exhibiting travelling wave behaviour initiated from a single nexus of infected cells at position $x = 0$. The infection travels an equal distance between successive times, demonstrating constant speed of spread. (*b*) A representative simulation of the PDE model with cell protection included ($\pi \neq 0$) showing how IFN signalling can stop the spread of infection by depleting target cells. The distribution of virions and infected cells can be seen to remain localized to the left side of the domain, as the rapid depletion of susceptible cells (as a result of increases in refractory cell population) in the domain prevent the infection from establishing a travelling wave. (Online version in colour.)

supplementary material, table S1 for the parameter values used for simulation) to compare the viral dynamics with and without paracrine IFN signalling (figure 1*b*). In agreement with the analytical derivation for $R_0$, we found that IFN paracrine signalling has negligible impact on the viral load during initial exponential growth period. This is true even for very large (biologically unrealistic) values of $\pi$ (electronic supplementary material, figure S1). We found that IFN-mediated protection of target cells is only able to affect the course of infection after some period of viral growth once infected cell concentration, and thus IFN concentration, rises to a sufficiently high level that there is a notable impact on protecting target cells and infected cells. The peak viral load is decreased by approximately $1/(1 + \pi)$-fold and the time to peak viraemia is relatively insensitive to changes in $\pi$ (see analytical approximations in electronic supplementary material). This nominal decrease in the time to peak viraemia is a consequence of the accelerated target cell depletion due to IFN signalling to uninfected cells.

Overall, our results show that when cells, viruses and IFNs are well-mixed (no spatial segregation is considered), autocrine signalling may have limited impact on the infection dynamics when a small fraction of cells turn on an antiviral response, and paracrine signalling has no impact on the infection dynamics during early infection.

## (b) A spatio-temporal PDE model of viral infection with IFN response

For almost all respiratory and enteric viral infections, the site of initial infection and viral replication is epithelial tissue, which is characterized by a monolayer structure [26]. Due to local diffusion of viral progeny over the epithelium, a virion is highly likely to infect one of a small number of neighbouring cells rather than having an equal probability of infecting any target cell, as is the implicit assumption in an ODE model of viral infection.

To incorporate the spatial structure of host cells, we constructed a PDE model (see equation (2.2) and Methods). We then simulated the PDE model with and without IFN signalling (figure 2; see electronic supplementary material, table S2 for the parameter values used for simulation). In the absence of the signalling ($\pi = k = f = 0$; figure 2*a*), the solution of the PDE model exhibits a travelling wave solution (called the infection wave below). Analysing the PDE model, we found that a front of infected cells propagates through healthy epithelium with a constant velocity, $v^*$ (see Methods). An approximate expression for $v^*$ is as follows (see electronic supplementary material):

$$v^* \approx \sqrt{\frac{D_V}{4}\left(\sqrt{108 T_0 \beta p} - 9c\right)}. \tag{3.1}$$

This expression shows that the spread of infection is driven primarily by the production ($p$) and diffusion ($D_V$) of virions, the infection of target cells ($\beta$) and the density of available target cells leading the front of the infection wave, i.e. $T_0$. The IFN signalling has the effect of both decreasing the production of virions ($p$) and decreasing the number of cells susceptible to infection ($T_0$), and thus it can in principal slow the spread of infection.

We then simulated the model with or without the autocrine and/or the paracrine signalling. With the inclusion of IFN paracrine signalling, we find that target cells at the front of infection are more likely to become refractory than infected due to the high diffusivity of IFN relative to virions (figures 2*b* and 3*a*). This causes target cell density leading the front of infected cells to decrease as the number of refractory cells rises. As the infection continues to spread, the IFN level becomes high enough leading to the depletion

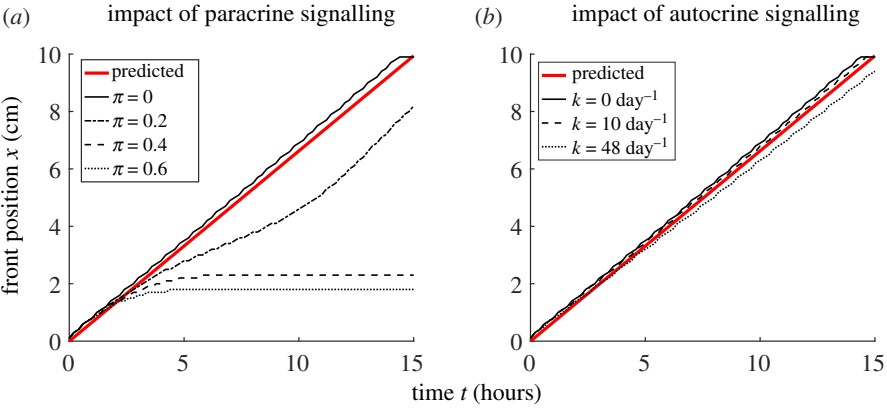

**Figure 3.** Comparison of the impacts of paracrine signalling and autocrine signalling on viral spatial spread. Shown is the position $x(t)$ of the infection front over time for various values of the IFN production parameter $\pi$ while keeping $k = 0$ and the autocrine-mediated transition rate $k$ while keeping $\pi = 0$. Here, we define the front position $x(t)$ to be the position that 1% of total cells are infected, i.e. $I(x(t), t) = 0.01T_0$. (a) Sufficiently strong paracrine signalling, e.g. setting $\pi = 10$ and $k = 0$, leads to halting the spread of infectious front. (b) Strong autocrine signalling, e.g. setting $\pi = 0$ and $k = 8$ day$^{-1}$, impact modestly on the speed of the viral spread. The red lines show the predicted front position given the analytically derived wave speed in equation (3.1). (Online version in colour.)

of target cells ($T_0$ in equation ((3.1))), which in turn impedes the spread of infection. Therefore, the PDE model predicts that paracrine signalling can have a strong impact on the spatial spread of virus infection by protecting cells at the front of infection.

In the absence of paracrine IFN signalling (figure 3b), we found that surprisingly, the observed travelling wave speed of the infection does not depend strongly on the strength of autocrine signalling, i.e. the value of the autocrine parameter $k$. This is because the speed of spread is mostly driven by virus production from cells at the wavefront. These infected cells are unlikely to be in an antiviral state, because of the waiting time (on average $1/k$ days) for that to occur. Thus, the results from the PDE model is in a sharp contrast to the results form the ODE model, with respect to the roles of the autocrine and the paracrine signalling on preventing the growth of infection than paracrine signalling.

## (c) A stochastic cellular automata model of the IFN response to viral infection

The analysis of the PDE model above identified important roles of IFN paracrine signalling on the spatial and temporal dynamics of virus infection. However, the deterministic nature of the PDE model neglects the inherent stochasticity present in the early stages of viral infection by considering a continuous real-valued density of virus and IFNs rather than individual particles. Furthermore, being a parabolic system of PDEs, the densities of virus and IFN, once produced, become instantaneously non-zero everywhere in the domain. Thus, the PDE model predicts that virus infection continues in locations far away from the initial site of infection over long period of time irrespective of how strong the IFN response is (electronic supplementary material, figure S2).

Here, to understand the spread of infection in the presence of IFN signalling in a more realistic setting, we constructed a 2D cellular automaton (CA) model, similar to previous works [32,39]. See Methods and electronic supplementary material for detailed description.

We first simulated the CA model assuming there is no IFN produced (figure 4a). In this case, the area of infection spreads radially. In the absence of IFN, the number of infected cells increases roughly quadratically with time, suggesting that the

speed of infection spread is constant (number of cells infected can be approximated by the area of infection $\pi r^2 = \pi (v^* t)^2$), which agrees with our analysis of the PDE model. We find the growth to be sub-quadratic in the presence of effective IFN signalling, which we observe as the decrease in the slope in the log–log plots of figure 4 as IFN production increases. When IFN particles are produced at a low level, the infection spreads roughly the same distance as the `ifn_prod = 0` case (`ifn_prod = 1` in figure 4b), though a lot of cells are protected by the IFN paracrine signalling (compare log–log plots in figure 4b with figure 4a). When IFN production increases further (`ifn_prod = 5`), target cells at the boundary of the front are more likely to be protected by IFN binding before becoming infected, leading to irregular spread of infection that depends heavily on virions diffusing a large distance before contacting a susceptible cell. Thus, virus spread becomes more stochastic as the spread of infection depends on rarer and rarer events (figure 4b). When IFN production is sufficiently large (`ifn_prod = 20`; figure 4d), all target cells near to infected cells are rapidly protected, leaving the virions produced each time step very unlikely to find a susceptible cell. In this way, an insulating layer of protected cells makes the continued spread of infection highly unlikely. The stronger the amount of IFN production, the thicker this layer of protected cells becomes, decreasing the probability of a virion reaching the healthy cells on the other side.

The results of the CA model corroborate our fundamental observations from the PDE model—that infection spreads at a constant rate when no IFN is present and that IFN signalling can slow and stop the spread of infection by decreasing the availability of susceptible cells in areas that are close to the site of infection. This pattern is reminiscent of the 'ring vaccination' strategy in epidemiological control [29–31] where an infectious disease outbreak can be effectively controlled by vaccinating those individuals who are close to or highly likely to contact infected individuals. Furthermore, the CA model exhibits stochastic behaviours that are inadmissible in deterministic models, highlighting infection establishment during initial infection may be a stochastic event. Overall, we find that by modelling the infection and immune signalling process as spatially dependent reveals how IFN signalling can halt the spread of infection on short time and length scales by isolating infectious units from susceptible target cells.

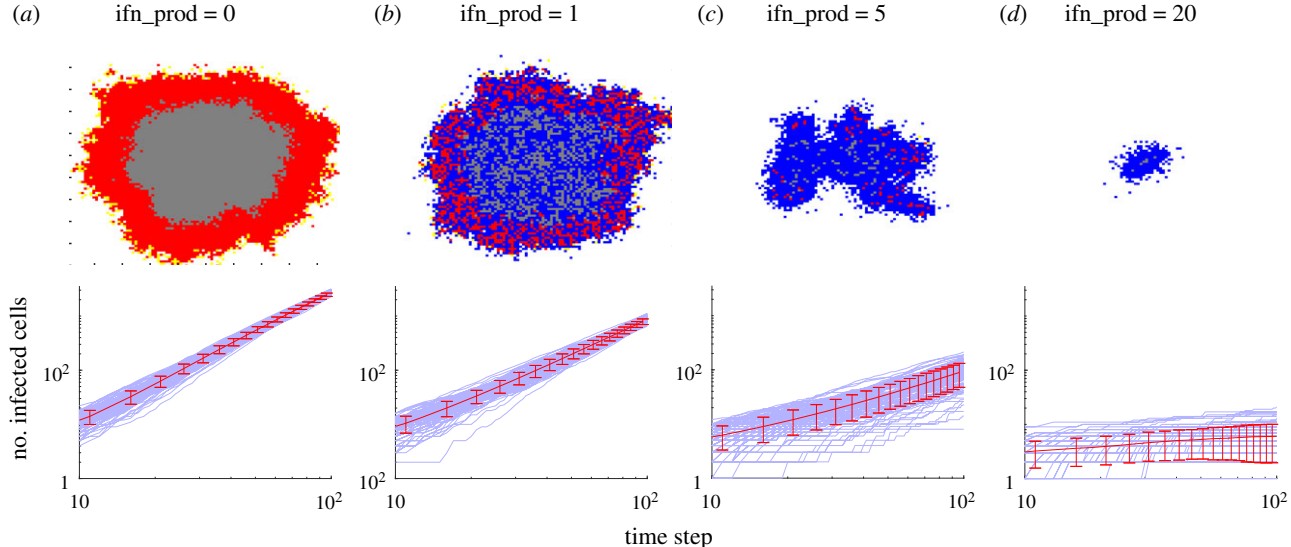

**Figure 4.** High production of IFN (`ifn_prod`) interferes with or stops the spread of infection by insulating infected cells from target cells with a layer of refractory cells. Top row shows example of simulations of the cellular automata (CA) model at 100 time steps on a $100 \times 100$ grid. Bottom row shows log-log plots of the number infected cells verses time for 100 independent simulations of CA with mean and standard deviation plotted in red. The approximate linear trend shown on the log-log plots suggests that the number of infected cells fits roughly a power law in time, $I(t) \propto t^\gamma$, where $\gamma$ is the slope of the trend line. For `ifn_prod` = 0 and 1, $\gamma$ is approximately 2, consistent with a travelling wave solution of the PDE model. $\gamma$ decreases as `ifn_prod` increases. In the images of the CA grid, red, blue and grey dots indicate infected cells, refractory/protected cells and dead cells, respectively; Target cells are left white. (Online version in colour.)

## 4. Discussion

Using a variety of models, we demonstrated here the mechanisms by which the autocrine and the paracrine IFN signalling stop an infection before it becomes systematic. Particularly, we showed that when there exists spatial structure in host target cells (a likely scenario especially in epithelium during initial infection), the IFN response can halt an infection by rapidly inducing an anti-viral state in susceptible cells close to infected cells, thus inhibiting the ability of the infection to spread. This is probably one important mechanism by which IFN signalling is effective in suppressing early infections in epithelial tissues. These results may have important implications towards understanding the impact of early IFN response on viral dynamics and its long-term role in defining disease outcomes of acute infections, such as SARS-CoV-2 infection [13]. Furthermore, as we argue below, the way the IFN response controls virus infection in a host is reminiscent of the 'ring vaccination' and 'contact tracing' strategies in epidemiological control. Quantitative understandings of the innate immune response may provide new insights for developing effective control strategies at the epidemiological scale.

Our work clarifies the roles of autocrine and paracrine signalling and quantifies their impacts on the initiation and spread of viral infection in different host-cell environments. When there exists a strong host-cell spatial structure where virion and IFN activities are restricted to locations near to where they are produced as considered in the PDE and CA models, the impact of paracrine signalling in shaping the progression of spreading infection becomes remarkably strong due to its ability to act locally. This is a likely scenario for initial stage of infection where the number of infected cells are low and infection often occurs in a restricted area of tissue epithelial cells. However, when infection occurs in an environment where spatial structure of host cells is not important, such as during a later stage of respiratory

infections where the virus population is already very large and immune response become systematic or during infections in the blood where host cells move around and contact each other, our ODE model predicts the role of autocrine signalling to be much more important than that of paracrine signalling in stymieing viral growth during early infection. Therefore, our work suggests that the two mechanisms of IFN signalling are probably complementary to one another, though one may be more or less impactful than the other depending on the context of the infection.

Previous modelling work using ODEs predicted that the innate immune response has a strong impact on viral dynamics close to or after peak viraemia [22,24,25]. Potent IFN response decreases the peak viral load, and decreases in IFN levels and consequently increases in target cell numbers are important to explain viral load dynamics after viral peak. Here, by explicitly considering the spatial structure of viral spread and the innate immune response, we show that potent innate immune response is able to strongly act on viral spread to slow down or even stop the virus spread during early infection. This again highlights the important role of the IFN response throughout the infection before adaptive immune response is developed. Recently, it was hypothesized that early stochastic events in viral mutation and innate immune response during influenza infection may have long-term impact on infection outcomes and disease severity [40]. In SARS-CoV-2 infection, the development of severe disease symptoms is likely due to suppression of the early antiviral response mediated by IFNs and consequently excess production of proinflammatory cytokines [13]. We believe that the model framework we proposed here, together with well-mixed approaches (such as in [22–25]), will be crucial to test the role of and quantify the impact of the IFN response during early acute infections, such as influenza and SARS-CoV-2, and how that may impact on disease outcome.

Our conclusions about how IFN response stops spatial viral spread are consistent with many lines of experimental

observations. For example, it is shown in [41] that paracrine IFN signalling was able to arrest the spread of infection in a monolayer by a rapid induction of downstream immune factors in proximal cells. Another example comes from chronic HCV infection of the liver, where the infection is highly spatially inhomogeneous, exhibiting clusters of infected hepatocytes surrounded by uninfected cells in which expressions of the IFN stimulated genes are high [42]. This emphasizes that IFN signalling could play an important role in the segregation of HCV-positive cells into localized clusters, preventing further spread by protecting cells in a neighbourhood of the cluster [43]. In another work, *in vitro* experiments have shown that IFN-suppressing wild-type vesicular stomatitis virus (VSV) is out-competed by mutants lacking IFN-suppression when either host-cell IFN response or monolayer spatial structure is removed. However, when monolayer spatial structure is preserved, the IFN-suppressing phenotype emerges as the dominant strain universally [44], emphasizing the importance of spatial structure in determining the effectiveness of the IFN signalling.

The IFN response to virus spread among cells at the host level has clear parallels in infectious disease transmission among individuals at the epidemiological level. First, autocrine signalling has an epidemiological parallel to testing and self-isolation in epidemiological control, where infectious individuals self-isolate in response to becoming aware of their own infection status through testing. In both cases, an individual cell or person's infectivity is modulated in response to the discovery of their own infection status. Second, paracrine signalling is in a clear analogy to contact tracing and quarantine, where the aim is to trace the at-risk individuals who contacted the infected individual and reduce the risk of further transmission [45]. Third, when the viral spread is mostly spatial, we showed that the collective host response through IFN diffusion leads to an outer layer of protected cells to isolate the infected cells from other susceptible cells.

This is a pattern reminiscent of ring vaccination or ring culling [29–31]. Fourth, IFNs signalling to neighbouring cells is also similar to the spread of disease awareness at the epidemiological scale. As analysed in [46], information about a disease spreads to those close to infected individuals in a contact network, and thus decreases the susceptibility of the informed to infection, suppressing the spread of the disease.

Overall, given that the IFN response is a highly effective immediate response employed by host cells in a wide variety of tissues and body compartments [8,36], we reason that it is likely to be a highly effective and robust strategy to prevent virus spread in a host, irrespective of the molecular details of the infection. At the epidemiological level, interventions discussed above (i.e. testing, isolation, contact tracing, ring vaccination/culling as well as spread of awareness) are likely to be effective and robust strategies against the spread of infectious diseases, although their relative effectiveness may depend on how the pathogen spreads through a population. Altogether, further experimental and modelling works on a quantitative understanding of the IFN response against virus infection will continue to offer new insights into virus infection, treatment and control at both the within-host level and the population level.

Data accessibility. The data and materials are included in the main text and the electronic supplementary material of the manuscript.
Authors' contributions. G.M.L., B.S. and R.K. conceived the project. G.M.L., H.R. and R.K. developed models. G.M.L. and H.R. performed analysis. G.M.L., B.S. and R.K. wrote the paper.
Competing interests. We declare we have no competing interests.
Funding. R.K. is funded by the DARPA INTERCEPT program (W911NF-17-2-0034). G.M.L. received support from the Research Training Group in Mathematical Biology, funded by a National Science Foundation grant (RTG/DMS–1246991). B.S. is funded by National Institutes of Health grant (NIH AI083333.).
Acknowledgements. We thank Katia Koelle, Rob de Boer, Christopher Brooke and members of the Ke laboratory for helpful discussions.

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
