## [Peer Review File · Proceedings of the Royal Society B: Biological Sciences]

Review History

RSPB-2020-3002.R0 (Original submission)

Review form: Reviewer 1

Recommendation

Accept with minor revision (please list in comments)

Scientific importance: Is the manuscript an original and important contribution to its field?

Good

General interest: Is the paper of sufficient general interest?

Good

Quality of the paper: Is the overall quality of the paper suitable?

Excellent

Is the length of the paper justified?

Yes

Should the paper be seen by a specialist statistical reviewer?

No

Do you have any concerns about statistical analyses in this paper? If so, please specify them explicitly in your report.

No

It is a condition of publication that authors make their supporting data, code and materials available - either as supplementary material or hosted in an external repository. Please rate, if applicable, the supporting data on the following criteria.

Is it accessible?

No

Is it clear?

N/A

Is it adequate?

No

Do you have any ethical concerns with this paper?

No

Comments to the Author

I appreciate the thoughtful responses of the authors to the original round of comments. I am still very enthusiastic about this interesting paper and the revisions definitely helped clarify things I was previously confused about.

I wonder if the code for the simulation results (especially the CA model) could/should be made available?

Apart from that, I noticed a few, extremely minor typographical things.

- l. 36 should "systematic" be "systemic"?
- l. 86 add "of" in "(b) A Spatial Model Viral Infection..."
- l. 129, is there a period missing here after "occurs"?
- l. 136, missing end parenthesis ")" after "[4,35]"?
- l. 454 "equilibrbium"-> "equilibrium"
- l. 488, don't forget to drop "borth"! :)

This is a really nice contribution.

Review form: Reviewer 2 (Rustom Antia)

Recommendation

Accept as is

Scientific importance: Is the manuscript an original and important contribution to its field?

Excellent

General interest: Is the paper of sufficient general interest?

Excellent

Quality of the paper: Is the overall quality of the paper suitable?

Excellent

Is the length of the paper justified?

Yes

Should the paper be seen by a specialist statistical reviewer?

No

Do you have any concerns about statistical analyses in this paper? If so, please specify them explicitly in your report.

No

It is a condition of publication that authors make their supporting data, code and materials available - either as supplementary material or hosted in an external repository. Please rate, if applicable, the supporting data on the following criteria.

Is it accessible?

N/A

Is it clear?

N/A

Is it adequate?

N/A

Do you have any ethical concerns with this paper?

No

Comments to the Author

The authors have addressed the issues I raised in my earlier review. This is one of the most interesting papers that I have read in a while. I enthusiastically recommend publication. What would be very nice to see (potentially in a follow up by the authors), is a more detailed investigation of the parameter regimes where the conventional ODE models fail (i.e. give very different results from the spatial models with interferon diffusion). However that should be left for later and this paper is a super first step.

Decision letter (RSPB-2020-3002.R0)

18-Jan-2021

Dear Dr Ke

I am pleased to inform you that your manuscript RSPB-2020-3002 entitled "Autocrine and paracrine interferon signaling as "ring vaccination" and "contact tracing" strategies to suppress virus infection in a host" has been accepted for publication in Proceedings B.

The referees have recommended publication, but also suggest some minor revisions to your manuscript. Therefore, I invite you to respond to the referees' comments and revise your manuscript. Because the schedule for publication is very tight, it is a condition of publication that you submit the revised version of your manuscript within 7 days. If you do not think you will be able to meet this date please let us know.

To revise your manuscript, log into <https://mc.manuscriptcentral.com/prsb> and enter your Author Centre, where you will find your manuscript title listed under "Manuscripts with Decisions." Under "Actions," click on "Create a Revision." Your manuscript number has been

appended to denote a revision. You will be unable to make your revisions on the originally submitted version of the manuscript. Instead, revise your manuscript and upload a new version through your Author Centre.

It is a condition of publication that data supporting your paper are made available either in the electronic supplementary material or through an appropriate repository. Please see our Data Sharing Policies <https://royalsociety.org/journals/authors/author-guidelines/#data>.

If you wish to submit your data to Dryad (<http://datadryad.org/>) and have not already done so you can submit your data via this link [http://datadryad.org/submit?journalID=RSPB&manu=\(Document not available\)](http://datadryad.org/submit?journalID=RSPB&manu=(Document+not+available)) which will take you to your unique entry in the Dryad repository. If you have already submitted your data to dryad you can make any necessary revisions to your dataset by following the above link. Please see <https://royalsociety.org/journals/ethics-policies/data-sharing-mining/> for more details.

Sincerely,
 Professor Hans Heesterbeek
 mailto: proceedingsb@royalsociety.org

Reviewer(s)' Comments to Author:

Referee: 1

Comments to the Author(s)

I appreciate the thoughtful responses of the authors to the original round of comments. I am still very enthusiastic about this interesting paper and the revisions definitely helped clarify things I was previously confused about.

I wonder if the code for the simulation results (especially the CA model) could/should be made available?

Apart from that, I noticed a few, extremely minor typographical things.

- l. 36 should "systematic" be "systemic"?
- l. 86 add "of" in "(b) A Spatial Model Viral Infection..."
- l. 129, is there a period missing here after "occurs"?
- l. 136, missing end parenthesis ")" after "[4,35]"?
- l. 454 "equilirbium" -> "equilibrium"
- l. 488, don't forget to drop "borth"! :)

This is a really nice contribution.

Referee: 2

Comments to the Author(s)

The authors have addressed the issues I raised in my earlier review. This is one of the most interesting papers that I have read in a while. I enthusiastically recommend publication. What would be very nice to see (potentially in a follow up by the authors), is a more detailed investigation of the parameter regimes where the conventional ODE models fail (i.e. give very different results from the spatial models with interferon diffusion). However that should be left for later and this paper is a super first step.

Decision letter (RSPB-2020-3002.R1)

26-Jan-2021

Dear Dr Ke

I am pleased to inform you that your manuscript entitled "Autocrine and paracrine interferon signaling as "ring vaccination" and "contact tracing" strategies to suppress virus infection in a host" has been accepted for publication in Proceedings B.

Open Access

Paper charges

Sincerely,

Proceedings B
